# Association between mixed venous oxygen saturation and serum uric acid levels in patients with heart failure

Yuto Mashitani◯*◉, Kazuo Ogawa◯◉, Ryuji Funaki, Yoshiro Tanaka, Takuya Oh, Toshikazu D. Tanaka◯, Tomohisa Nagoshi◯, Kosuke Minai◯, Makoto Kawai, Michihiro Yoshimura◯

Department of Internal Medicine, Division of Cardiology, The Jikei University School of Medicine, Tokyo, Japan

◉ These authors contributed equally to this work.
* h20ms-mashitani@jikei.ac.jp

**Data Availability Statement:** The data sets used in this study contains personal information of patients at the Jikei University Hospital, to which we belong, and is managed by the Ethics Committee of The

## Abstract

Hypoxia leads to increased purine metabolism in tissues, resulting in increased serum uric acid levels, and may also cause impaired uric acid excretion in the kidneys and intestinal tract. However, the relationship between hypoxia and serum uric acid levels in patients with heart failure remains largely unexplored. Because mixed venous oxygen saturation is an acute indicator of systemic oxygenation, in this study, we investigated the relationship between mixed venous oxygen saturation and serum uric acid levels. This retrospective analysis included 386 patients with heart failure who underwent cardiac catheterization at our institution. The relationship between mixed venous oxygen saturation and serum uric acid levels was examined by single regression analysis. Stratified regression analysis, structural equation modeling, and partial correlation analysis were used to examine the effects of eight factors known to influence mixed venous oxygen saturation and serum uric acid levels. The single regression analysis showed a significant negative correlation between mixed venous oxygen saturation and serum uric acid levels (*P*<0.001). Significant negative correlations were also observed in many subgroups in the stratified analysis, in the path diagram based on structural equation modeling, and in the partial correlation analysis. These results suggest that there may be a direct relationship between mixed venous oxygen saturation and serum uric acid levels that is not mediated by any known factor.

## Introduction

Hyperuricemia has been associated with hypertension, metabolic syndrome, dementia, as well as coronary artery, cerebrovascular, and kidney disease [1–7]. Prior studies have also reported associated risk factors for hyperuricemia, including age, sex, body mass index (BMI), estimated glomerular filtration rate (eGFR), triglyceride (TG) levels, and glycated hemoglobin (HbA1c) levels, but some differences exist among reports [8–13].

Hypoxia adversely affects various tissues in the body. Although its effects on nucleic acid synthesis have not been fully investigated, it is possible that the production of uric acid (UA), a

Jikei University School of Medicine for Biomedical Research. The data sets used in this study contains personal information of patients at the Jikei University Hospital, and in principle, its public release is restricted. These restrictions have been imposed by the Ethics Committee of The Jikei University School of Medicine for Biomedical Research. However, if you would like to obtain the data, please contact the Jikei University School of Medicine Biomedical Research Ethics Committee (contact via The Jikei University School of Medicine, Telephone number: +81 -3-3433-1111, Email address: rinri@jikei.ac.jp) and you will be able to obtain an anonymized dataset without any problems. In addition, this dataset is guaranteed to be stored for at least 10 years.

**Funding:** The author(s) received no specific funding for this work.

**Competing interests:** The authors have declared that no competing interests exist.

terminal metabolite of purines, is enhanced as energy metabolism is reduced [14]. In addition, hypoxia in the kidneys and intestinal tract may also result in impaired UA excretion [15–18]. Hence, hypoxia is likely to cause hyperuricemia via accelerated synthesis and decreased excretion of UA. Although there have been some reports showing the relationship between heart disease and serum UA levels, there have been few studies that have examined the relationship between hypoxia and serum UA levels in clinical settings, particularly in patients with heart failure [19–21]. Elevated UA levels have been reported in patients with sleep apnea syndrome [22–24]. Although these studies strongly suggest a relationship between hypoxia and hyperuricemia, they are limited to a specific patient population. In addition, a more sensitive index of tissue oxygenation may be needed to demonstrate the relationship between hypoxia and hyperuricemia more clearly.

Mixed venous oxygen saturation ($SvO_2$) is measured in blood samples from the pulmonary artery, containing mixed blood from the superior and inferior vena cava and coronary sinus, and represents the amount of oxygen remaining in venous blood [25,26]. Thus, $SvO_2$ may be an appropriate indicator of the degree of systemic oxygenation [27,28]. Theoretically, $SvO_2$ is related to arterial blood oxygen saturation (SaO2), oxygen consumption ($VO_2$), hemoglobin (Hb) levels, cardiac output (CO), and other factors [28,29]. Clinically, $SvO_2$ is a valuable indicator for guiding systemic management, particularly in critically ill patients. However, as $SvO_2$ measurement requires cardiac catheterization, which is an invasive procedure, not many clinical studies have investigated $SvO_2$. Our hypothesis is that while uric acid levels vary due to various factors, hypoxia is involved in hyperuricemia independently of the factors reported so far. The purpose of this study is to examine the relationship between hypoxia and uric acid levels in heart failure patients who underwent cardiac catheterization. As a methodology to demonstrate this, this study uses $SvO_2$ as an index of hypoxia and employs various statistical methods.

## Materials and methods

### Study population

We accessed our database on March 19, 2022 to begin our study. We retrospectively reviewed the records of patients with heart failure admitted to the Department of Cardiology of our hospital between June 2017 and May 2022. Patients who underwent cardiac catheterization for cardiac function evaluation and had $SvO_2$ measurements performed were included. Patients who received oxygen therapy, those with pulmonary arterial hypertension, and those with left or right shunts were excluded. Pulmonary arterial hypertension was defined as a mean pulmonary arterial pressure of $\geq$20 mmHg and pulmonary arterial wedge pressure of $\leq$15 mmHg, according to the guidelines on pulmonary hypertension.

This study was approved by the Ethics Committee of The Jikei University School of Medicine for Biomedical Research (study protocol: 24-355(7121)). We complied with the routine ethical regulations of our institution. All clinical investigations were conducted in accordance with the principles set forth in the Declaration of Helsinki. As this was a retrospective study, instead of obtaining informed consent from each patient, we posted a notice about the study design and contact information according to our routine ethical regulations on the official website of our institution(https://jikei.bvits.com/rinri/publish.aspx). In this public notification, we ensured that patients had the opportunity to refuse to participate (opt-out) in the study.

### Parameter measurement

$SvO_2$ measurement was performed using a Swan–Ganz thermodilution catheter (Bioptimal, Japan). The catheter was inserted through the femoral vein and its distal end was placed in the

main pulmonary artery under fluoroscopy. Blood was drawn from the distal end; the first 10 mL of blood collected were discarded, after which blood was collected using a blood gas syringe, and any residual air was discarded immediately. The blood sample was immediately analyzed with a blood gas analyzer (ABL800 FLEX, Radiometer, Denmark) to determine the $SvO_2$ value. CO was measured by thermodilution. Measurements were taken at least twice, and the average value was used as the measured value. Left ventricular catheterization was performed using a 4-F or 5-F Judkins right catheter (Terumo, Japan). Left ventricular ejection fraction was calculated by left ventriculography using a PIG-tail catheter (Terumo, Japan). In addition, various hematological and biochemical parameters were measured at the central laboratory of the hospital in peripheral venous blood samples obtained in the early morning on the day of catheterization after fasting.

## Statistical analysis

Continuous data were summarized as mean ± standard deviation or median [upper and lower quartiles] depending on their distribution. Categorical data were summarized as frequencies with percentages.

First, we performed single correlation analysis between $SvO_2$ and serum UA levels. Next, to eliminate as much as possible the effects of other factors on the relationship between $SvO_2$ and serum UA levels, we conducted stratified regression analysis. Based on previous studies, age, sex, BMI, eGFR, TG level, and HbA1c level may influence serum UA levels [8–13], while Hb level and CO (in the current study we used the CI) may influence $SvO_2$ [28,29]. Hence, in the stratified analysis, we examined whether the relationship between $SvO_2$ and serum UA levels would be altered by these eight factors by dividing the study population into two groups for each factor. For age, sex, BMI, CI, and TG, HbA1c, and Hb levels, the cutoff values were the median values in our population, and for eGFR, the cutoff value was 60 mL/min/1.73 m$^2$. We performed single regression analysis in each subgroup. Next, we devised path diagrams based on structural equation modeling to examine the relationship between $SvO_2$ and serum UA levels considering the possible effect of the above eight factors on $SvO_2$ and serum UA levels, as well as the effect of $SvO_2$ on serum UA levels. Finally, we conducted partial correlation analysis for the relationship between $SvO_2$ and serum UA levels with dialysis, urate-lowering drugs, diuretics, the above eight factors, and all these combined as control variables.

Statistical analysis was performed using IBM SPSS Statistics version 28.0 (IBM Corp, Armonk, NY, USA). Structural equation modeling was performed using IBM SPSS Amos version 28 (Amos Development Corporation, Meadville, PA, USA). A *P* value of less than 0.05 was considered to indicate statistical significance.

## Results

### Characteristics of the study population

A flowchart of the patient screening process is shown in Fig 1. A total of 386 patients (male: n = 264, 68.4%) with a mean age of 70.2 ± 14.3 years were included in the analysis. The patients' clinical characteristics are shown in Table 1. The mean $SvO_2$ was 67.2 ± 7.62% and the mean serum UA level was 6.43 ± 2.24 mg/dL. The median age, BMI, cardiac index (CI), and TG, HbA1c, and Hb levels were 73 years, 22.4 kg/m$^2$, 2.51 L/min/m$^2$, 93 mg/dL, 5.9%, and 12.4 g/dL, respectively.

### Single regression analysis results

The results of the single regression analysis showed a significant negative correlation between $SvO_2$ and serum UA levels (Table 2 and Fig 2).

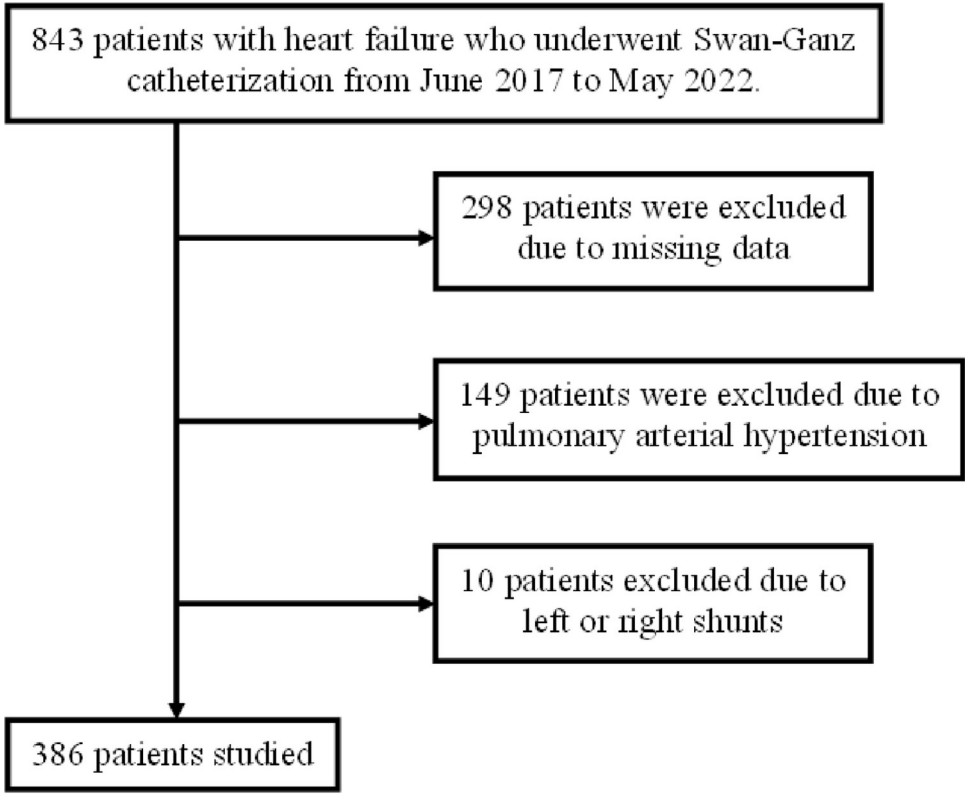

**Fig 1. Patient screening flowchart.**

### Stratified regression analysis results

In the regression analysis with stratification (Table 3), a significant negative correlation between $SvO_2$ and serum UA levels was observed in all subgroups except in those with eGFR <60 mL/min/1.73 m$^2$, HbA1c >5.9%, CI ≥2.51 L/min/m$^2$, CI <2.51 L/min/m$^2$, and Hb <12.4 g/dL.

### Structural equation modeling results

The path diagram based on structural equation modeling is shown in Fig 3. Detailed results of the structural equation modeling are presented in Table 4. We observed a significant negative correlation between $SvO_2$ and serum UA levels, with a standardized estimate of -0.055 ($P = 0.003$).

### Partial correlation analysis results

Table 5 shows the results of the partial correlation analysis. Significant negative relationships between $SvO_2$ and serum UA levels were observed with dialysis ($P = 0.003$), urate-lowering drugs ($P<0.001$), diuretics ($P = 0.008$), $SvO_2$ and all eight factors ($P<0.001$), and all factors combined ($P = 0.005$) as control variables.

## Discussion

In this study, we observed a significant negative relationship between $SvO_2$ and serum UA levels, independent of all known factors (age, sex, BMI, eGFR, CI, TG, HbA1c, and Hb levels,

**Table 1. Clinical characteristics of the study population.**

| Characteristics (N = 386) | Number (%) or mean ± SD | Median (IQR) |
|---|---|---|
| Sex (male/female) | 264/122 (68.4/31.6) | |
| Age (years) | 70.2±14.3 | 73 (62–81) |
| BMI (kg/m$^2$) | 22.9±4.43 | 22.4 (20.1–24.9) |
| Active tobacco smoking | 57(14.8) | 12.4 (10.7–14.125) |
| Hb (g/dL) | 12.5±2.33 | 1.03 (0.80–1.53) |
| Cr (mg/dL) | 2.10±2.70 | 51.4 (31.7–66.9) |
| eGFR (mL/min/1.73 m$^2$) | 50.1±28.8 | 6.2 (4.8–7.7) |
| UA (mg/dL) | 6.43±2.24 | 103 (90–124) |
| FBS (mg/dL) | 113.5±38.1 | 5.8 (5.5–6.3) |
| HbA1c (%) | 6.05±0.91 | 93 (71–129) |
| TG (mg/dL) | 103.7±45.6 | 50 (41–62) |
| HDL-C (mg/dL) | 52.8±16.1 | 98 (75–123.8) |
| LDL-C (mg/dL) | 101.2±33.3 | 1.9 (1.4–2.6) |
| LDL-C/HDL-C | 2.06±0.83 | 0.21 (0.07–0.77) |
| CRP (mg/dL) | 0.97±2.25 | 258.8 (106.3–526.2) |
| BNP (pg/mL) | 468.9±614.1 | 42.3 (32.3–53.7) |
| LVEF (%) | 43.5±15.2 | 2.51 (2.06–3.01) |
| CI (L/min/m$^2$) | 2.6±0.68 | 67.6 (63.1–72.4) |
| SvO$_2$ (%) | 67.2±7.62 | |
| Underlying disease | | |
| Atrial fibrillation | 48 (12.4) | |
| Hypertension | 286 (74.1) | |
| Diabetes mellitus | 126 (32.6) | |
| Dyslipidemia | 197 (51.0) | |
| Renal dysfunction* | 241 (62.4) | |
| Hyperuricemia* | 224 (58.0) | |
| Hemodialysis | 44 (11.4) | |
| Ischemic heart disease | 149 (38.6) | |
| Valvular heart disease | 137 (35.5) | |
| Severe TR | 5 (1.30) | |
| Cardiomyopathy | 103 (26.7) | |
| Constrictive pericarditis | 6 (1.55) | |
| Myocarditis | 4 (1.04) | |
| Congenital heart disease | 3 (0.78) | |
| Medications | | |
| ACE inhibitors | 121 (31.3) | |
| ARBs | 105 (27.2) | |
| Beta blockers | 250 (64.8) | |
| Calcium channel blockers | 127 (32.9) | |
| Diuretics | 246 (63.7) | |
| Statins | 157 (40.7) | |
| Non-statin for dyslipidemia | 58 (15.0) | |
| Oral antidiabetic agents | 66 (17.1) | |
| Insulin | 24 (6.22) | |
| GLP-1 receptor agonist | 14 (3.63) | |
| UA-lowering agents | 130 (33.7) | |
| SGLT2 inhibitor | 24 (6.22) | |

*Renal dysfunction = eGFR < 60 mL/min/1.73 m$^2$.

*Hyperuricemia = UA >7.0 mg/dL or UA-lowering drug users.

ACE, angiotensin-converting enzyme; ARB, angiotensin receptor blocker; BNP, brain natriuretic peptide; CI, cardiac index; Cr, creatinine; CRP, C-reactive protein; eGFR, estimated glomerular filtration rate; FBS, fasting blood sugar; GLP, glucagon-like peptide; HbA1c, glycated hemoglobin; HDL-C, high-density cholesterol; LDL-C, low-density cholesterol; LVEF, left ventricular ejection fraction; SD, standard deviation; SGLT, sodium–glucose cotransporter.

dialysis, urate-lowering drugs, and diuretics) that may affect these two parameters. These results suggest that the relationship between SvO$_2$ and serum UA levels may be direct and not mediated by known factors.

**Table 2. Results of the single regression analysis of SvO₂ and UA levels.**

| Groups | Non-standardized coefficient | | Standardized regression coefficient | Test statistic | P value | 95% CI |
|---|---|---|---|---|---|---|
| | Regression coefficient | Standard error | | | | |
| All patients | -0.050 | 0.015 | -0.170 | -3.375 | <0.001 | -0.079 to -0.021 |

CI, confidence interval; SvO₂, mixed venous oxygen saturation; UA, uric acid.

The significant negative relationship between SvO₂ and serum UA levels was observed in all analyses except in several subgroups in the stratified analysis. Although these results do not completely rule out the possibility that other factors influence this relationship, it can be said that there is a relationship between SvO₂ and serum UA levels. One possible reason for the lack of significant correlation between these parameters in the stratified analysis in the high HbA1c, low eGFR, low Hb, and both CI subgroups is the small number of patients in these subgroups. However, other possible reasons should also be considered.

### Stratified analysis of SvO₂ and serum uric acid levels

**Impact of HbA1c, eGFR, hemoglobin, and cardiac index.** The levels of SvO₂ and UA were not correlated in the high HbA1c subgroup. In insulin-resistant states, such as metabolic syndrome, hyperinsulinemia increases the protein expression of urate transporter 1 (URAT1) in the proximal tubules, leading to an enhanced uptake of UA [30,31]. Conversely, in diabetes, the increase in glucose concentration in the renal tubules activates the glucose transporter 9 (GLUT9) isoform responsible for uric acid excretion in the proximal tubules [32]. The results of this study may be attributed to the likelihood that UA levels were relatively lower in the high

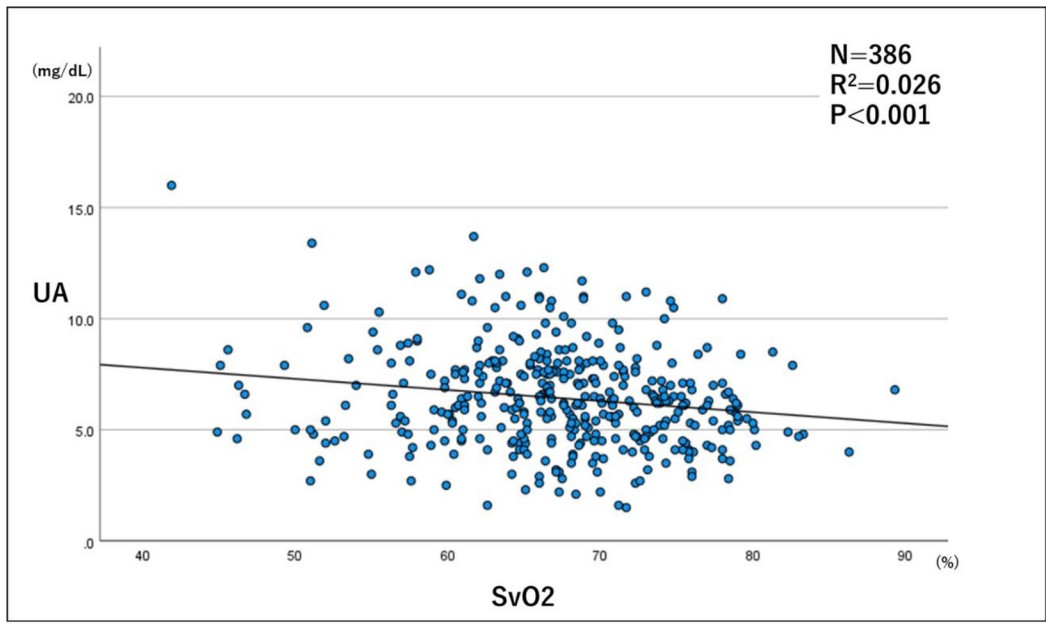

**Fig 2. Relationship between SvO2 and serum UA levels in the whole study population.** A scatterplot showing the relationship between SvO2 and serum UA levels based on the single regression analysis with SvO2 as the independent variable and serum UA level as the dependent variable. As a correlation was found between UA and SvO2 levels, a regression line is drawn. N, number of samples; R2, coefficient of determination; SvO2, mixed venous oxygen saturation; UA, uric acid.

Table 3. Results of the stratified regression analysis.

| | Non-standardized coefficient | | Standardized regression coefficient | Test statistic | P value | 95% CI |
|---|---|---|---|---|---|---|
| | Regression coefficient | Standard error | | | | |
| Classification by age | | | | | | |
| ≥73 y.o. | -0.062 | 0.022 | -0.195 | -2.787 | 0.006 | -0.106 to -0.018 |
| <73 y.o. | -0.041 | 0.020 | -0.148 | -2.041 | 0.043 | -0.080 to -0.001 |
| Classification by sex | | | | | | |
| Male | -0.041 | 0.018 | -0.140 | -2.289 | 0.023 | -0.077 to -0.006 |
| Female | -0.076 | 0.025 | -0.264 | -3.004 | 0.003 | -0.126 to -0.026 |
| Classification by BMI | | | | | | |
| ≥22.4 | -0.062 | 0.022 | -0.199 | -2.834 | 0.005 | -0.105 to -0.019 |
| <22.4 | -0.045 | 0.020 | -0.158 | -2.222 | 0.027 | -0.085 to -0.005 |
| Classification by eGFR | | | | | | |
| ≥60 | -0.066 | 0.023 | -0.237 | -2.920 | 0.004 | -0.111 to -0.021 |
| <60 | -0.038 | 0.020 | -0.125 | -1.948 | 0.053 | -0.077 to 0.000 |
| Classification by TG level | | | | | | |
| ≥93 | -0.042 | 0.021 | -0.142 | -1.983 | 0.049 | -0.084 to 0.000 |
| <93 | -0.056 | 0.021 | -0.195 | -2.725 | 0.007 | -0.097 to -0.015 |
| Classification by HbA1c level | | | | | | |
| ≥5.9% | -0.021 | 0.021 | -0.074 | -1.008 | 0.315 | -0.063 to 0.020 |
| <5.9% | -0.075 | 0.021 | -0.244 | -3.497 | <0.001 | -0.117 to -0.033 |
| Classification by CI | | | | | | |
| ≥2.51 | -0.019 | 0.022 | -0.062 | -0.853 | 0.395 | -0.063 to -0.025 |
| <2.51 | -0.043 | 0.024 | -0.129 | -1.780 | 0.077 | -0.091 to -0.005 |
| Classification by Hb level | | | | | | |
| ≥12.4 | -0.095 | 0.021 | -0.308 | -4.502 | <0.001 | -0.137 to -0.054 |
| <12.4 | -0.035 | 0.021 | -0.121 | -1.665 | 0.098 | -0.076 to 0.006 |

BMI, body mass index; CI, cardiac index; eGFR, estimated glomerular filtration rate; Hb, hemoglobin; HbA1c, glycated hemoglobin; TG, triglyceride.

HbA1c group, presumably due to the latter mechanism. There is no apparent contradiction in considering these mechanisms as related, as supported by findings in other studies [11,33,34].

SvO$_2$ and UA levels were also not associated in the low eGFR subgroup. Although the relationship between renal dysfunction and hyperuricemia is well known, various detailed mechanisms have been proposed. One mechanism of renal impairment induced by hyperuricemia is due to the effect of monosodium urate monohydrate crystals in the kidneys, as well as in joints and other organs, which activate the NLRP3 inflammasome cascade and lead to interleukin-1β activation [35]. Another proposed mechanism is that hyperuricemia induces intracellular oxidative stress, endothelial dysfunction, renal fibrosis, and glomerulosclerotic effects [36,37]. These mechanisms may result in impaired UA excretion, and other renal factors may also cause fluctuations in serum UA levels, making the relationship between SvO$_2$ and serum UA levels less apparent.

SvO$_2$ was not associated with UA levels in the low Hb subgroup. A prior study has reported a positive correlation between serum UA and iron and ferritin levels [38]. Hence, the relationship between SvO$_2$ and serum UA levels in this subgroup may be less prominent because iron deficiency may reduce xanthine oxidoreductase (XOR) activity and UA levels [39,40].

The lack of association between SvO$_2$ and UA levels in both the low and high CI subgroups suggests that the low CI subgroup had a small number of patients, whereas the high CI

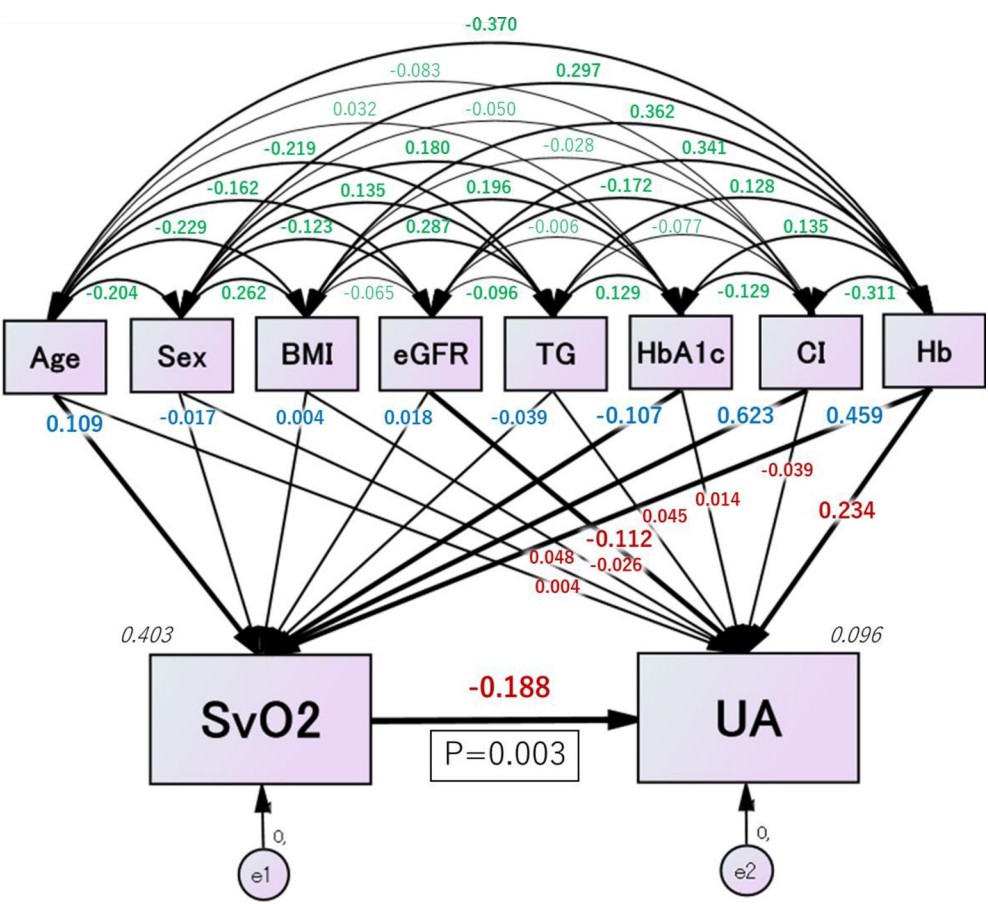

**Fig 3. Path diagram based on structural equation modeling (for all patients).** The path diagram is devised from structural equations among eight factors affecting SvO2, UA levels, and both SvO2 and UA levels and also shows the effects of SvO2 on UA levels. The paths between variables are represented by one-way arrows extending from the independent to the dependent variable, indicating a positive or negative effect, whereas two-way arrows between the two variables indicate a correlation. Dependent variables are accompanied by error variables (e), one-way arrows by estimates of the standardized coefficient (blue & red), and two-way arrows by estimates of the correlation coefficient (green). Squared multiple correlations are shown in narrow italics. SvO2 shows a significant relationship with UA levels (P<0.01). BMI, body mass index; CI, cardiac index; eGFR, estimated glomerular filtration rate; Hb, hemoglobin; HbA1c, glycated hemoglobin; SvO2, mixed venous oxygen saturation; TG, triglyceride; UA, uric acid; e, extraneous variable.

subgroup had a generally higher $SvO_2$ due to sufficient CO, which may have diminished the relationship with UA levels.

In other words, the present results for HbA1c, eGFR, Hb, and CI may mask the relationship between $SvO_2$ and serum UA levels but do not negate this relationship, as each of these factors can be explained.

## Mechanisms of hyperuricemia induced by hypoxic conditions

**Synthesis and excretion of uric acid.** Hypoxic conditions may lead to hyperuricemia by promoting UA synthesis and reducing its excretion. The mechanisms involved are as follows.

**Promotion of uric acid synthesis under hypoxic conditions.** In a hypoxic state, insufficient oxygen supply results in decreased ATP production and increased ATP breakdown. This process leads to the breakdown of adenine nucleotides (such as ATP, ADP, and AMP), which increases hypoxanthine levels. XOR then converts hypoxanthine and xanthine into uric acid.

**Table 4. Structural equation modeling results.**

| Clinical factor | | | Estimate | Standard error | Test statistic | P value | Standard estimate | | |
|---|---|---|---|---|---|---|---|---|---|
| | | | | | | | Direct effect | Indirect effect | Total effect |
| SvO$_2$ (R$^2$ = 0.403) | ← | Age | 0.058 | 0.024 | 2.398 | 0.016 | 0.109 | 0 | 0.109 |
| | ← | Sex | -0.277 | 0.716 | -0.387 | 0.699 | -0.017 | 0 | -0.017 |
| | ← | BMI | 0.006 | 0.078 | 0.079 | 0.937 | 0.004 | 0 | 0.004 |
| | ← | eGFR | 0.005 | 0.012 | 0.402 | 0.688 | 0.018 | 0 | 0.018 |
| | ← | TG | -0.007 | 0.007 | -0.914 | 0.361 | -0.039 | 0 | -0.039 |
| | ← | HbA1c | -0.900 | 0.348 | -2.583 | 0.010 | -0.107 | 0 | -0.107 |
| | ← | CI | 6.918 | 0.481 | 14.382 | <0.001 | 0.623 | 0 | 0.623 |
| | ← | Hb | 1.500 | 0.171 | 8.769 | <0.001 | 0.459 | 0 | 0.459 |
| UA (R$^2$ = 0.096) | ← | Age | 0.001 | 0.009 | 0.077 | 0.938 | 0.004 | -0.020 | -0.016 |
| | ← | Sex | 0.234 | 0.259 | 0.905 | 0.366 | 0.048 | 0.003 | 0.051 |
| | ← | BMI | -0.013 | 0.028 | -0.466 | 0.641 | -0.026 | -0.001 | -0.027 |
| | ← | eGFR | -0.009 | 0.004 | -2.047 | 0.041 | -0.112 | -0.003 | -0.115 |
| | ← | TG | 0.002 | 0.003 | 0.857 | 0.391 | 0.045 | 0.007 | 0.052 |
| | ← | HbA1c | 0.035 | 0.127 | 0.272 | 0.786 | 0.014 | 0.020 | 0.034 |
| | ← | CI | -0.128 | 0.217 | -0.587 | 0.557 | -0.039 | -0.117 | -0.156 |
| | ← | Hb | 0.226 | 0.068 | 3.322 | <0.001 | 0.234 | -0.086 | 0.148 |
| | ← | SvO$_2$ | -0.055 | 0.019 | -2.984 | 0.003 | -0.188 | 0 | -0.188 |

BMI, body mass index; CI, cardiac index; eGFR, estimated glomerular filtration rate; Hb, hemoglobin; HbA1c, glycated hemoglobin; SvO$_2$, mixed venous oxygen saturation; TG, triglyceride; UA, uric acid.

Additionally, hypoxic conditions increase oxidative stress, causing damage to DNA and cell membranes. This damage accelerates the breakdown of nucleic acids and further increases purine metabolites. The heightened oxidative stress enhances XOR activation, thereby promoting uric acid synthesis [14,41,42].

**Sites of uric acid synthesis.** The specific organs or sites where uric acid synthesis occurs under hypoxic conditions are still debated. While it is not yet clear where uric acid is specifically generated, it is possible that uric acid synthesis is promoted throughout the body. Among them, adipose tissue is an important site for uric acid synthesis [43,44]. Hypoxanthine secretion from human adipose tissue increases under hypoxic conditions. This hypoxanthine may be converted into uric acid by XOR in other tissues, such as the liver or endothelial cells. Recent research indicates that in the liver, hypoxia-inducible factor 1α (HIF-1α), activated by fatty acid oxidation, increases the expression of NT5C2 and XDH, thereby promoting

**Table 5. Partial correlation analysis with all possible affecting factors as control factors.**

| Groups | Control variable | Independent variable | Partial correlation coefficient | P value |
|---|---|---|---|---|
| All patients | HD | SvO$_2$<−>UA | -0.139 | 0.003 |
| | Urate-lowering drugs | | -0.173 | <0.001 |
| | Diuretics | | -0.122 | 0.008 |
| | Age, sex, BMI, eGFR, TG, HbA1c, CI, Hb | | -0.168 | <0.001 |
| | Age, sex, BMI, eGFR, TG, HbA1c, CI, Hb, HD, urate-lowering drugs, diuretics | | -0.134 | 0.005 |

BMI, body mass index; CI, cardiac index; eGFR, estimated glomerular filtration rate; HD, hemodialysis; Hb, hemoglobin; HbA1c, glycated hemoglobin; SvO$_2$, mixed venous oxygen saturation; TG, triglyceride; UA, uric acid.

hypoxanthine transport and uric acid synthesis [45]. Furthermore, a decrease in $SvO_2$ may also adversely affect lung tissue. This is because the pulmonary arteries are partially responsible for supplying nutrients to the lungs. Given this, it is plausible that hypoxia-induced damage to lung tissue could potentially promote uric acid synthesis.

**Hypoxia and uric acid excretion.**   The mechanisms underlying the relationship between hypoxia and decreased UA excretion are not yet fully understood. Approximately two-thirds of UA in the human body is excreted by the kidneys, while the remaining one-third is excreted by the intestine. Hypoxemia may suppress uric acid excretion by affecting desmin protein levels in podocytes and Na+-K+-ATPase activity [18]. In the current study, low $SvO_2$ levels were found to correlate with low CI levels, suggesting that reduced CI might partially contribute to renal dysfunction and decreased urine production, potentially leading to decreased UA excretion [46]. Further research is needed to explore the detailed impact of hypoxia on UA excretion in urine.

**Intestinal uric acid excretion.**   In the intestine, sirtuin-1 (SIRT1) accelerates UA excretion by activating the ATP-binding cassette transporter G2 (ABCG2) [16,17]. Since SIRT1 activity is inhibited by hypoxia [15], low oxygen levels in the small intestine might lead to SIRT1 inactivation and impaired UA excretion due to the inactivation of ABCG2.

## Limitations

This study has some limitations. First, the retrospective study design may have introduced selection bias and affects the generalizability of our findings. Second, the study was conducted at a single institution and our sample size was relatively small, which may have limited the statistical power to detect significant differences between groups in the structural equation modeling analysis. However, the clear demonstration of a relationship between $SvO_2$ and serum UA levels even in a limited number of patients suggests that there may be a direct relationship between $SvO_2$ and serum UA levels. Third, serum UA levels are affected not only by increased production and impaired excretion of UA but also by the amount of purine ingested in the diet. Although the patients in this study fasted prior to cardiac catheterization, the influence of daily diet cannot be completely ruled out. Fourth, $SvO_2$ is theoretically related to $SaO_2$, $VO_2$, Hb levels, and CO. In the present study, data on $SaO_2$ and $VO_2$ were not available and were not included in the analysis. In particular, $SaO_2$ may be related to UA levels. Although this is a subject for a future study, it is expected that $SaO_2$ will show considerably higher values than $SvO_2$, and the use of $SvO_2$ seems more appropriate for the main purpose of this study. Finally, although the results of this study suggest that $SvO_2$ and serum UA levels are closely related, it is not clear whether a truly direct relationship can be said to exist. We merely showed a relationship and cannot rule out the possibility of other influencing factors. Continued exploration of factors affecting serum UA levels is needed. To overcome the limitations of this study, it is important to conduct multi-center studies with a larger and more diverse patient population. A prospective study design should be used to minimize selection bias and control variables more strictly. Detailed dietary records should be kept to consider the influence of purine intake, and additional data on $SaO_2$, $VO_2$, Hb levels, and CO should be collected and analyzed. Longitudinal studies should be conducted to observe changes over time, and multivariate analysis or machine learning techniques should be used to account for confounding factors. Basic research using animal models is also necessary to provide a more detailed understanding of the relationship between $SvO_2$ and serum UA levels.

## Conclusion

$SvO_2$ and serum UA levels are factors influenced by various factors. We studied the relationship between $SvO_2$ and serum UA using stratified analysis, partial correlation analysis, and

structural equation modeling to take into account factors influencing $SvO_2$ and serum UA levels. Our results suggest that there may be a direct relationship between $SvO_2$ and serum UA levels in heart failure patients that is not mediated by known factors. However, we only demonstrated a relationship and could not completely exclude the possibility of other influencing factors. Therefore, further studies are needed to evaluate the relationship between $SvO_2$ and serum UA levels.

## Acknowledgments

We thank all participating physicians and nurses for their contributions to this study. We would also like to thank ELSEVIER Language Editing Services for English language editing.

## Author Contributions

**Conceptualization:** Yuto Mashitani.

**Data curation:** Yuto Mashitani.

**Formal analysis:** Yuto Mashitani.

**Investigation:** Yuto Mashitani, Kosuke Minai.

**Methodology:** Yuto Mashitani.

**Project administration:** Yuto Mashitani.

**Supervision:** Kazuo Ogawa, Ryuji Funaki, Yoshiro Tanaka, Takuya Oh, Toshikazu D. Tanaka, Tomohisa Nagoshi, Kosuke Minai, Makoto Kawai, Michihiro Yoshimura.

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
