## [Decision Letter · Decision Letter 0]

2 Aug 2024

PONE-D-24-22570Association between mixed venous oxygen saturation and serum uric acid levels in patients with heart failurePLOS ONE

Dear Dr. Mashitani,

Thank you for submitting your manuscript to PLOS ONE. After careful consideration, we feel that it has merit but does not fully meet PLOS ONE’s publication criteria as it currently stands. Therefore, we invite you to submit a revised version of the manuscript that addresses the points raised during the review process.

We look forward to receiving your revised manuscript.

Kind regards,

Sepiso Kenias Masenga, PhD

Academic Editor

PLOS ONE

Journal Requirements:

Reviewers' comments:

Reviewer's Responses to Questions

**Comments to the Author**

1. Is the manuscript technically sound, and do the data support the conclusions?

Reviewer #1: Yes

Reviewer #2: Yes

2. Has the statistical analysis been performed appropriately and rigorously? 

Reviewer #1: Yes

Reviewer #2: Yes

3. Have the authors made all data underlying the findings in their manuscript fully available?

Reviewer #1: Yes

Reviewer #2: Yes

4. Is the manuscript presented in an intelligible fashion and written in standard English?

Reviewer #1: Yes

Reviewer #2: Yes

5. Review Comments to the Author

**Reviewer #1: **Manuscript/paper title: Association between mixed venous oxygen saturation and serum uric acid levels in patients with heart failure

Summary statement of the article:

This is an insightful article and I extend my applause for this piece of work and the effort the authors put into conducting such research. The study investigated the relationship between mixed venous oxygen saturation (SvO2) and serum uric acid (UA) levels in patients with heart failure. The analysis included 386 patients and used various statistical methods to determine a significant negative correlation between SvO2 and UA levels where authors suggested that, there was a direct relationship but not mediated by known factors.

Specific areas of improvement

Major comments:

1. Clarify Hypothesis and Objectives: The introduction should clearly state the hypothesis and specific objectives of the study. This will help readers understand the purpose and significance of the research. Line # 38-65.

2. Expand on Mechanisms: The discussion section should provide more detailed explanations of the biological or immunological mechanisms underlying the observed relationship between SvO2 and UA levels. This will strengthen the interpretation of the results. Line # 180-247.

3. Address Limitations: While the limitations have been mentioned, it would be beneficial to discuss how these limitations might impact the study’s findings and suggest ways to address them in future research.

Minor Corrections:

1. Authors could consider adding a subheading for text under Limitations. Line # 248.

2. Figures and Tables: Authors should ensure that all figures and tables fit at least on one page to avoid overlapping. Their narrations should also be close. This will help readers follow the data presentation more easily and clearly. Line # 132-139, 156-157, 164-168, and 177-178.

3. References: Verify that all references are correctly formatted and up-to-date. This will enhance the credibility of the manuscript. “Hence, hypoxia is likely to cause hyperuricemia via accelerated synthesis and decreased excretion of UA. Nonetheless, few studies have examined the relationship between hypoxia and serum UA levels in clinical settings, 50 particularly in patients with heart failure.” Lines # 48-50 please add citations.

• “Mixed venous oxygen saturation (SvO2) is measured in blood samples from the pulmonary artery, containing mixed blood from the superior and inferior vena cava and coronary sinus, and represents the amount of oxygen remaining in venous blood.” Line # 55-57. Please provide citations and do the same for other statements that may need to be cited throughout the text.

**Reviewer #2: **Article Title

Association between mixed venous oxygen saturation and serum uric acid levels in patients with heart failure.

Summary

The study by Mashitani et al. explores the relationship between mixed venous oxygen saturation (SvO2) and serum uric acid (UA) levels in 386 heart failure patients who underwent cardiac catheterization. The authors performed regression analyses and structural equation modeling to assess the relationship. Results indicated a significant negative correlation between SvO2 and serum UA levels, suggesting a direct relationship not mediated by known factors like age, sex, BMI, eGFR, and HbA1c levels.

Major Strengths of Study

The authors did a comprehensive analysis which enhanced the validity of the study findings. The study gives novel insights into the direct relationship between systemic oxygenation and uric acid levels in heart failure patients, thus addressing critical research gaps. Additionally, the article is well-written.

Despite the above strengths, the authors need to address the following issues to improve the quality of their work.

Comments

1. In line 130 the authors mention that the median Cardiac Index (CI) is 2.52 L/min/m2, however in Table 1, the median is 2.51. The inconsistency also spills into the stratified regression analysis where the text reads, "for CI, the cutoff values were the median values in our population," yet later the cutoff is referred to as "≥2.51" and "<2.51." The authors need to correct this inconsistency.

2. Severe TR in Table 1 has an absolute count of 5, in a sample size of 386 this would be around 1.3%, however, it is captured as 3.65%. Let the authors clarify this.

6. PLOS authors have the option to publish the peer review history of their article (what does this mean?). If published, this will include your full peer review and any attached files.

Reviewer #1: **Yes: **Bislom C. Mweene

Reviewer #2: **Yes: **Lweendo Muchaili

---

## [Author Response · Author response to Decision Letter 0]

26 Sep 2024

A point-by-point response to issues raised by referees.

Thank you very much for giving us excellent advice. These comments, suggestions and points of advice have helped us to improve the quality of our manuscript. We sincerely appreciate your taking the time to review our manuscript.

Before responding to the journal's request and the reviewer's comments, we have made a correction to one co-author's name.

Dr. Tanaka Toshikazu has a middle name, so I have revised his name.

The parts modified in the revised manuscript are shown in red text.

To the journal requirements

Comment; When submitting your revision, we need you to address these additional requirements.

Response: Thank you very much for your comment. I have revised my manuscript to meet PLOS ONE's style requirements based on the two PDFs you provided.

Comment; We note that you have indicated that there are restrictions to data sharing for this study. For studies involving human research participant data or other sensitive data, we encourage authors to share de-identified or anonymized data. However, when data cannot be publicly shared for ethical reasons, we allow authors to make their data sets available upon request. For information on unacceptable data access restrictions, please see http://journals.plos.org/plosone/s/data-availability#loc-unacceptable-data-access-restrictions. 

Response: Thank you very much for your very important comment. The data sets used in this study contains personal information of patients at the Jikei University Hospital, to which we belong, and is managed by the Ethics Committee of The Jikei University School of Medicine for Biomedical Research. The data sets used in this study contains personal information of patients at the Jikei University Hospital, and in principle, its public release is restricted. These restrictions have been imposed by the Ethics Committee of The Jikei University School of Medicine for Biomedical Research. However, if you would like to obtain the data, please contact the Jikei University School of Medicine Biomedical Research Ethics Committee and you will be able to obtain an anonymized dataset without any problems. In addition, this dataset is guaranteed to be stored for at least 10 years. We have updated the data availability statement on the submission form.

Comment; When completing the data availability statement of the submission form, you indicated that you will make your data available on acceptance. We strongly recommend all authors decide on a data sharing plan before acceptance, as the process can be lengthy and hold up publication timelines. Please note that, though access restrictions are acceptable now, your entire data will need to be made freely accessible if your manuscript is accepted for publication. This policy applies to all data except where public deposition would breach compliance with the protocol approved by your research ethics board. If you are unable to adhere to our open data policy, please kindly revise your statement to explain your reasoning and we will seek the editor's input on an exemption. Please be assured that, once you have provided your new statement, the assessment of your exemption will not hold up the peer review process.

Response: Thank you very much for your important comments. We would like to ask your journal for your support in releasing the data. Therefore, we have corrected 'Additional data availability information'.

Comment; Please review your reference list to ensure that it is complete and correct. If you have cited papers that have been retracted, please include the rationale for doing so in the manuscript text, or remove these references and replace them with relevant current references. Any changes to the reference list should be mentioned in the rebuttal letter that accompanies your revised manuscript. If you need to cite a retracted article, indicate the article’s retracted status in the References list and also include a citation and full reference for the retraction notice.

Response: Thank you very much for your comment. I have reviewed my reference list again. I have confirmed that I have not cited any retracted papers in this study. I have also revised my reference list in conjunction with the revision of my manuscript. Details are mentioned in my response to the reviewer's comments. The revisions to the reference list are noted in red text in the reference list in the 'Revised Manuscript with Track Changes' file.

To Reviewer #1’s Comments

Comment; Clarify Hypothesis and Objectives: The introduction should clearly state the hypothesis and specific objectives of the study. This will help readers understand the purpose and significance of the research. Line # 38-65.

Response: Thank you very much for your very important comment. We agree that it is very important to clearly state the hypothesis and specific objectives of the study in the introduction. Therefore, we have clearly stated the hypothesis and specific objectives of this study at the end of the introduction.

(New text in the Introduction)

Our hypothesis is that while uric acid levels vary due to various factors, hypoxia is involved in hyperuricemia independently of the factors reported so far. The purpose of this study is to examine the relationship between hypoxia and uric acid levels in heart failure patients who underwent cardiac catheterization. As a methodology to demonstrate this, this study uses SvO2 as an index of hypoxia and employs various statistical methods.

Comment; Expand on Mechanisms: The discussion section should provide more detailed explanations of the biological or immunological mechanisms underlying the observed relationship between SvO2 and UA levels. This will strengthen the interpretation of the results. Line # 180-247.

Response: Thank you very much for your very important and valuable comments.

As you pointed out, our manuscript seems to lack an explanation of the biological and immunological mechanisms of the relationship between hypoxia and serum uric acid levels.

Accordingly, we have changed the structure of the discussion text.

We added subtitles to the discussion and divided it into two parts: Stratified Analysis of SvO2 and Serum Uric Acid Levels and Mechanisms of Hyperuricemia Induced by Hypoxic Conditions.

In addition, in Mechanisms of Hyperuricemia Induced by Hypoxic Conditions, we divided the discussion into Promotion of Uric Acid Synthesis Under Hypoxic Conditions, Sites of Uric Acid Synthesis, and Hypoxia and Uric Acid Excretion.

Along with revising the text, we have added references to more reliably support our theory.

(New text in the Discussion)

Stratified Analysis of SvO2 and Serum Uric Acid Levels: Impact of HbA1c, eGFR, Hemoglobin, and Cardiac Index

The levels of SvO2 and UA were not correlated in the high HbA1c subgroup. In insulin-resistant states, such as metabolic syndrome, hyperinsulinemia increases the protein expression of urate transporter 1 (URAT1) in the proximal tubules, leading to an enhanced uptake of UA. (29, 30) Conversely, in diabetes, the increase in glucose concentration in the renal tubules activates the glucose transporter 9 (GLUT9) isoform responsible for uric acid excretion in the proximal tubules. (31) The results of this study may be attributed to the likelihood that UA levels were relatively lower in the high HbA1c group, presumably due to the latter mechanism. There is no apparent contradiction in considering these mechanisms as related, as supported by findings in other studies. (11, 32, 33) 

SvO2 and UA levels were also not associated in the low eGFR subgroup. Although the relationship between renal dysfunction and hyperuricemia is well known, various detailed mechanisms have been proposed. One mechanism of renal impairment induced by hyperuricemia is due to the effect of monosodium urate monohydrate crystals in the kidneys, as well as in joints and other organs, which activate the NLRP3 inflammasome cascade and lead to interleukin-1β activation.(34) Another proposed mechanism is that hyperuricemia induces intracellular oxidative stress, endothelial dysfunction, renal fibrosis, and glomerulosclerotic effects.(35, 36) These mechanisms may result in impaired UA excretion, and other renal factors may also cause fluctuations in serum UA levels, making the relationship between SvO2 and serum UA levels less apparent. 

SvO2 was not associated with UA levels in the low Hb subgroup. A prior study has reported a positive correlation between serum UA and iron and ferritin levels.(37) Hence, the relationship between SvO2 and serum UA levels in this subgroup may be less prominent because iron deficiency may reduce xanthine oxidoreductase (XOR) activity and UA levels.(38, 39) 

The lack of association between SvO2 and UA levels in both the low and high CI subgroups suggests that the low CI subgroup had a small number of patients, whereas the high CI subgroup had a generally higher SvO2 due to sufficient CO, which may have diminished the relationship with UA levels. 

In other words, the present results for HbA1c, eGFR, Hb, and CI may mask the relationship between SvO2 and serum UA levels but do not negate this relationship, as each of these factors can be explained.

Mechanisms of Hyperuricemia Induced by Hypoxic Conditions: Synthesis and Excretion of Uric Acid

Hypoxic conditions may lead to hyperuricemia by promoting UA synthesis and reducing its excretion. The mechanisms involved are as follows.

Promotion of Uric Acid Synthesis Under Hypoxic Conditions: In a hypoxic state, insufficient oxygen supply results in decreased ATP production and increased ATP breakdown. This process leads to the breakdown of adenine nucleotides (such as ATP, ADP, and AMP), which increases hypoxanthine levels. XOR then converts hypoxanthine and xanthine into uric acid. Additionally, hypoxic conditions increase oxidative stress, causing damage to DNA and cell membranes. This damage accelerates the breakdown of nucleic acids and further increases purine metabolites. The heightened oxidative stress enhances XOR activation, thereby promoting uric acid synthesis. (14, 40-44) 

Sites of Uric Acid Synthesis: The specific organs or sites where uric acid synthesis occurs under hypoxic conditions are still debated. While it is not yet clear where uric acid is specifically generated, it is possible that uric acid synthesis is promoted throughout the body. Among them, adipose tissue is an important site for uric acid synthesis. (45, 46) Hypoxanthine secretion from human adipose tissue increases under hypoxic conditions. This hypoxanthine may be converted into uric acid by XOR in other tissues, such as the liver or endothelial cells. Recent research indicates that in the liver, hypoxia-inducible factor 1α (HIF-1α), activated by fatty acid oxidation, increases the expression of NT5C2 and XDH, thereby promoting hypoxanthine transport and uric acid synthesis. (47) Furthermore, a decrease in SvO2 may also adversely affect lung tissue. This is because the pulmonary arteries are partially responsible for supplying nutrients to the lungs. Given this, it is plausible that hypoxia-induced damage to lung tissue could potentially promote uric acid synthesis

Hypoxia and Uric Acid Excretion: The mechanisms underlying the relationship between hypoxia and decreased UA excretion are not yet fully understood. Approximately two-thirds of UA in the human body is excreted by the kidneys, while the remaining one-third is excreted by the intestine. Hypoxemia may suppress uric acid excretion by affecting desmin protein levels in podocytes and Na+-K+-ATPase activity. (15) In the current study, low SvO2 levels were found to correlate with low CI levels, suggesting that reduced CI might partially contribute to renal dysfunction and decreased urine production, potentially leading to decreased UA excretion. (48) Further research is needed to explore the detailed impact of hypoxia on UA excretion in urine.

Intestinal Uric Acid Excretion: In the intestine, sirtuin-1 (SIRT1) accelerates UA excretion by activating the ATP-binding cassette transporter G2 (ABCG2). (16, 17) Since SIRT1 activity is inhibited by hypoxia (18), low oxygen levels in the small intestine might lead to SIRT1 inactivation and impaired UA excretion due to the inactivation of ABCG2.

Comment; Address Limitations: While the limitations have been mentioned, it would be beneficial to discuss how these limitations might impact the study’s findings and suggest ways to address them in future research.

Response: Thank you very much for your very valuable comments. I have added a paragraph to the discussion section about how to deal with the limitations of this study and future prospects.

(New text in the Discussion)

To overcome the limitations of this study, it is important to conduct multi-center studies with a larger and more diverse patient population. A prospective study design should be used to minimize selection bias and control variables more strictly. Detailed dietary records should be kept to consider the influence of purine intake, and additional data on SaO2, VO2, Hb levels, and CO should be collected and analyzed. Longitudinal studies should be conducted to observe changes over time, and multivariate analysis or machine learning techniques should be used to account for confounding factors. Basic research using animal models is also necessary to provide a more detailed understanding of the relationship between SvO2 and serum UA levels.

Comment; Authors could consider adding a subheading for text under Limitations. Line # 248.

Response: Thank you very much for your suggestion. We have added the subheading "Limitations."

Comment; Figures and Tables: Authors should ensure that all figures and tables fit at least on one page to avoid overlapping. Their narrations should also be close. This will help readers follow the data presentation more easily and clearly. Line # 132-139, 156-157, 164-168, and 177-178.

Response: Thank you very much for your important comment. I have arranged the tables and figures so that they can be viewed within one page.

Comment; References: Verify that all references are correctly formatted and up-to-date. This will enhance the credibility of the manuscript. “Hence, hypoxia is likely to cause hyperuricemia via accelerated synthesis and decreased excretion of UA. Nonetheless, few studies have examined the relationship between hypoxia and serum UA levels in clinical settings, 50 particularly in patients with heart failure.” Lines # 48-50 please add citations.

• “Mixed venous oxygen saturation (SvO2) is measured in blood samples from the pulmonary artery, containing 

---

## [Decision Letter · Decision Letter 1]

13 Nov 2024

Association between mixed venous oxygen saturation and serum uric acid levels in patients with heart failure

PONE-D-24-22570R1

Dear Dr. Mashitani,

We’re pleased to inform you that your manuscript has been judged scientifically suitable for publication and will be formally accepted for publication once it meets all outstanding technical requirements.

Kind regards,

Sepiso K. Masenga, PhD

Academic Editor

PLOS ONE

Additional Editor Comments (optional):

Reviewers' comments:

Reviewer's Responses to Questions

**Comments to the Author**

1. If the authors have adequately addressed your comments raised in a previous round of review and you feel that this manuscript is now acceptable for publication, you may indicate that here to bypass the “Comments to the Author” section, enter your conflict of interest statement in the “Confidential to Editor” section, and submit your "Accept" recommendation.

Reviewer #1: All comments have been addressed

Reviewer #2: All comments have been addressed

2. Is the manuscript technically sound, and do the data support the conclusions?

Reviewer #1: Yes

Reviewer #2: (No Response)

3. Has the statistical analysis been performed appropriately and rigorously? 

Reviewer #1: Yes

Reviewer #2: (No Response)

4. Have the authors made all data underlying the findings in their manuscript fully available?

Reviewer #1: Yes

Reviewer #2: (No Response)

5. Is the manuscript presented in an intelligible fashion and written in standard English?

Reviewer #1: Yes

Reviewer #2: (No Response)

6. Review Comments to the Author

Reviewer #1: All the concerns and comments that were noted have fully been attended to. It is great piece of work that authors have drafted and quite insightful. Just one recommendation to authors, to rearrange the manuscript to PLOSONE's formatting guidelines.

Reviewer #2: (No Response)

7. PLOS authors have the option to publish the peer review history of their article (what does this mean?). If published, this will include your full peer review and any attached files.

Reviewer #1: No

Reviewer #2: No

---

## [Editor Report · Acceptance letter]

18 Nov 2024

PONE-D-24-22570R1 

PLOS ONE

Dear Dr. Mashitani, 

I'm pleased to inform you that your manuscript has been deemed suitable for publication in PLOS ONE. Congratulations! Your manuscript is now being handed over to our production team.

Kind regards, 

on behalf of

Prof. Sepiso K. Masenga 

Academic Editor

PLOS ONE